# Bandit Learning in Concave $N$-Person Games

**Mario Bravo**
Universidad de Santiago de Chile
Departamento de Matemática y Ciencia de la Computación
mario.bravo.g@usach.cl

**David Leslie**
Lancaster University & PROWLER.io
d.leslie@lancaster.ac.uk

**Panayotis Mertikopoulos**
Univ. Grenoble Alpes, CNRS, Inria, Grenoble INP
LIG 38000 Grenoble, France.
panayotis.mertikopoulos@imag.fr

## Abstract

This paper examines the long-run behavior of learning with bandit feedback in non-cooperative concave games. The bandit framework accounts for extremely low-information environments where the agents may not even know they are playing a game; as such, the agents' most sensible choice in this setting would be to employ a no-regret learning algorithm. In general, this does not mean that the players' behavior stabilizes in the long run: no-regret learning may lead to cycles, even with perfect gradient information. However, if a standard monotonicity condition is satisfied, our analysis shows that no-regret learning based on mirror descent with bandit feedback converges to Nash equilibrium with probability $1$. We also derive an upper bound for the convergence rate of the process that nearly matches the best attainable rate for *single-agent* bandit stochastic optimization.

## 1 Introduction

The bane of decision-making in an unknown environment is *regret:* noone wants to realize in hindsight that the decision policy they employed was strictly inferior to a plain policy prescribing the same action throughout. For obvious reasons, this issue becomes considerably more intricate when the decision-maker is subject to situational uncertainty and the "fog of war": when the only information at the optimizer's disposal is the reward obtained from a given action (the so-called "bandit" framework), is it even possible to design a no-regret policy? Especially in the context of online convex optimization (repeated decision problems with continuous action sets and convex costs), this problem becomes even more challenging because the decision-maker typically needs to infer gradient information from the observation of a single scalar. Nonetheless, despite this extra degree of difficulty, this question has been shown to admit a positive answer: regret minimization *is* possible, even with bandit feedback (Flaxman et al., 2005; Kleinberg, 2004).

In this paper, we consider a multi-agent extension of this framework where, at each stage $n = 1, 2, \ldots,$ of a repeated decision process, the reward of an agent is determined by the actions of all agents via a fixed mechanism: *a non-cooperative $N$-person game.* In general, the agents – or players – might be completely oblivious to this mechanism, perhaps even ignoring its existence: for instance, when choosing how much to bid for a good in an online auction, an agent is typically unaware of who the other bidders are, what are their specific valuations, etc. Hence, lacking any knowledge about the game, it is only natural to assume that agents will at least seek to achieve a minimal worst-case guarantee and minimize their regret. As a result, a fundamental question that arises is *a*) whether the agents' sequence of actions stabilizes to a rationally admissible state under no-regret learning; and *b*) if it does, whether convergence is affected by the information available to the agents.

**Related work.**    In finite games, no-regret learning guarantees that the players' time-averaged, empirical frequency of play converges to the game's set of coarse correlated equilibria (CCE), and the rate of this convergence is $\mathcal{O}(1/n)$ for $(\lambda, \mu)$-smooth games (Foster et al., 2016; Syrgkanis et al., 2015). In general however, this set might contain highly subpar, rationally inadmissible strategies: for instance, Viossat and Zapechelnyuk (2013) provide examples of CCE that assign positive selection probability *only* to strictly dominated strategies. In the class of potential games, Cohen et al. (2017) recently showed that the *actual* sequence of play (i.e., the sequence of actions that determine the agents' rewards at each stage) converges under no-regret learning, even with bandit feedback. Outside this class however, the players' chosen actions may cycle in perpetuity, even in simple, two-player zero-sum games with full information (Mertikopoulos et al., 2018a,b); in fact, depending on the parameters of the players' learning process, agents could even exhibit a fully unpredictable, aperiodic and chaotic behavior (Palaiopanos et al., 2017). As such, without further assumptions in place, no-regret learning in a multi-agent setting does not necessarily imply convergence to a unilaterally stable, equilibrium state.

In the broader context of games with continuous action sets (the focal point of this paper), the long-run behavior of no-regret learning is significantly more challenging to analyze. In the case of mixed-strategy learning, Perkins and Leslie (2014) and Perkins et al. (2017) showed that mixed-stratgy learning based on stochastic fictitious play converges to an $\varepsilon$-perturbed Nash equilibrium in potential games (but may lead to as much as $\mathcal{O}(\varepsilon n)$ regret in the process). More relevant for our purposes is the analysis of Nesterov (2009) who showed that the time-averaged sequence of play induced by a no-regret dual averaging (DA) process with noisy gradient feedback converges to Nash equilibrium in monotone games (a class which, in turn, contains all concave potential games).

The closest antecedent to our approach is the recent work of Mertikopoulos and Zhou (2018) who showed that the *actual* sequence of play generated by dual averaging converges to Nash equilibrium in the class of variationally stable games (which includes all monotone games). To do so, the authors first showed that a naturally associated continuous-time dynamical system converges, and then used the so-called *asymptotic pseudotrajectory* (APT) framework of Benaïm (1999) to translate this result to discrete time. Similar APT techniques were also used in a very recent preprint by Bervoets et al. (2018) to establish the convergence of a *payoff-based* learning algorithm in two classes of one-dimensional concave games: games with strategic complements, and ordinal potential games with isolated equilibria. The algorithm of Bervoets et al. (2018) can be seen as a special case of mirror descent coupled with a two-point gradient estimation process, suggesting several interesting links with our paper.

**Our contributions.**    In this paper, we drop all feedback assumptions and we focus on the *bandit* framework where the only information at the players' disposal is the payoffs they receive at each stage. As we discussed above, this lack of information complicates matters considerably because players must now estimate their payoff gradients from their observed rewards. What makes matters even worse is that an agent may introduce a significant bias in the (concurrent) estimation process of another, so traditional, multiple-point estimation techniques for derivative-free optimization cannot be applied (at least, not without significant communication overhead between players).

To do away with player coordination requirements, we focus on learning processes which could be sensibly deployed in a single-agent setting and we show that, in monotone games, the sequence of play induced by a wide class of no-regret learning policies converges to Nash equilibrium with probability 1. Furthermore, by specializing to the class of strongly monotone games, we show that the rate of convergence is $\mathcal{O}(n^{-1/3})$, i.e., it is nearly optimal with respect to the attainable $\mathcal{O}(n^{-1/2})$ rate for bandit, *single-agent* stochastic optimization with strongly convex and smooth objectives (Agarwal et al., 2010; Shamir, 2013).

We are not aware of a similar Nash equilibrium convergence result for concave games with general convex action spaces and *bandit* feedback: the analysis of Mertikopoulos and Zhou (2018) requires first-order feedback, while the analysis of Bervoets et al. (2018) only applies to one-dimensional games. We find this outcome particularly appealing for practical applications of game theory (e.g., in network routing) because it shows that in a wide class of (possibly very complicated) nonlinear games, the Nash equilibrium prediction does not require full rationality, common knowledge of rationality, flawless execution, or even the knowledge that a game is being played: a commonly-used, individual no-regret algorithm suffices.

## 2 Problem setup and preliminaries

**Concave games.** Throughout this paper, we will focus on games with a finite number of players $i \in \mathcal{N} = \{1, \ldots, N\}$ and continuous action sets. During play, every player $i \in \mathcal{N}$ selects an *action* $x_i$ from a compact convex subset $\mathcal{X}_i$ of a $d_i$-dimensional normed space $\mathcal{V}_i$; subsequently, based on each player's individual objective and the *action profile* $x = (x_i; x_{-i}) \equiv (x_1, \ldots, x_N)$ of all players' actions, every player receives a *reward*, and the process repeats. In more detail, writing $\mathcal{X} \equiv \prod_i \mathcal{X}_i$ for the game's *action space*, we assume that each player's reward is determined by an associated *payoff* (or *utility*) *function* $u_i \colon \mathcal{X} \to \mathbb{R}$. Since players are not assumed to "know the game" (or even that they are involved in one) these payoff functions might be a priori unknown, especially with respect to the dependence on the actions of other players. Our only structural assumption for $u_i$ will be that $u_i(x_i; x_{-i})$ is concave in $x_i$ for all $x_{-i} \in \mathcal{X}_{-i} \equiv \prod_{j \neq i} \mathcal{X}_j$, $i \in \mathcal{N}$.

With all this in hand, a *concave game* will be a tuple $\mathcal{G} \equiv \mathcal{G}(\mathcal{N}, \mathcal{X}, u)$ with players, action spaces and payoffs defined as above. Below, we briefly discuss some examples thereof:

**Example 2.1** (Cournot competition). In the standard Cournot oligopoly model, there is a finite set of *firms* indexed by $i = 1, \ldots, N$, each supplying the market with a quantity $x_i \in [0, C_i]$ of some good (or service), up to the firm's production capacity $C_i$. By the law of supply and demand, the good is priced as a decreasing function $P(x_{\text{tot}})$ of the total amount $x_{\text{tot}} = \sum_{i=1}^{N} x_i$ supplied to the market, typically following a linear model of the form $P(x_{\text{tot}}) = a - b x_{\text{tot}}$ for positive constants $a, b > 0$. The utility of firm $i$ is then given by

$$u_i(x_i; x_{-i}) = x_i P(x_{\text{tot}}) - c_i x_i, \tag{2.1}$$

i.e., it comprises the total revenue from producing $x_i$ units of the good in question minus the associated production cost (in the above, $c_i > 0$ represents the marginal production cost of firm $i$).

**Example 2.2** (Resource allocation auctions). Consider a service provider with a number of splittable *resources* $s \in \mathcal{S} = \{1, \ldots, S\}$ (bandwidth, server time, GPU cores, etc.). These resources can be leased to a set of $N$ bidders (players) who can place monetary bids $x_{is} \geq 0$ for the utilization of each resource $s \in \mathcal{S}$ up to each player's total budget $b_i$, i.e., $\sum_{s \in \mathcal{S}} x_{is} \leq b_i$. Once all bids are in, resources are allocated proportionally to each player's bid, i.e., the $i$-th player gets $\rho_{is} = (q_s x_{is})/(c_s + \sum_{j \in \mathcal{N}} x_{js})$ units of the $s$-th resource (where $q_s$ denotes the available units of said resource and $c_s \geq 0$ is the "entry barrier" for bidding on it). A simple model for the utility of player $i$ is then given by

$$u_i(x_i; x_{-i}) = \sum_{s \in \mathcal{S}} [g_i \rho_{is} - x_{is}], \tag{2.2}$$

with $g_i$ denoting the marginal gain of player $i$ from acquiring a unit slice of resources.

For more examples of monotone games, see Scutari et al. (2010), D'Oro et al. (2015), Mertikopoulos and Belmega (2016), and references therein.

**Nash equilibrium and monotone games.** The most widely used solution concept for non-cooperative games is that of a *Nash equilibrium* (NE), defined here as any action profile $x^* \in \mathcal{X}$ that is resilient to unilateral deviations, viz.

$$u_i(x_i^*; x_{-i}^*) \geq u_i(x_i; x_{-i}^*) \quad \text{for all } x_i \in \mathcal{X}_i, i \in \mathcal{N}. \tag{NE}$$

By the classical existence theorem of Debreu (1952), every concave game admits a Nash equilibrium. Moreover, thanks to the individual concavity of the game's payoff functions, Nash equilibria can also be characterized via the first-order optimality condition

$$\langle v_i(x^*), x_i - x_i^* \rangle \leq 0 \quad \text{for all } x_i \in \mathcal{X}_i, \tag{2.3}$$

where $v_i(x)$ denotes the individual payoff gradient of the $i$-th player, i.e.,

$$v_i(x) = \nabla_i u_i(x_i; x_{-i}), \tag{2.4}$$

with $\nabla_i$ denoting differentiation with respect to $x_i$.[1] In terms of regularity, it will be convenient to assume that each $v_i$ is Lipschitz continuous; to streamline our presentation, this will be our standing assumption in what follows.

Starting with the seminal work of Rosen (1965), much of the literature on continuous games and their applications has focused on games that satisfy a condition known as *diagonal strict concavity* (DSC). In its simplest form, this condition posits that there exist positive constants $\lambda_i > 0$ such that

$$\sum_{i \in \mathcal{N}} \lambda_i \langle v_i(x') - v_i(x), x_i' - x_i \rangle < 0 \quad \text{for all } x, x' \in \mathcal{X}, \ x \neq x'. \tag{DSC}$$

Owing to the formal similarity between (DSC) and the various operator monotonicity conditions in optimization (see e.g., Bauschke and Combettes, 2017), games that satisfy (DSC) are commonly referred to as (strictly) *monotone*. As was shown by Rosen (1965, Theorem 2), monotone games admit a unique Nash equilibrium $x^* \in \mathcal{X}$, which, in view of (DSC) and (NE), is also the unique solution of the (weighted) variational inequality

$$\sum_{i \in \mathcal{N}} \lambda_i \langle v_i(x), x_i - x_i^* \rangle < 0 \quad \text{for all } x \neq x^*. \tag{VI}$$

This property of Nash equilibria of monotone games will play a crucial role in our analysis and we will use it freely in the rest of our paper.

In terms of applications, monotonicity gives rise to a very rich class of games. As we show in the paper's supplement, Examples 2.1 and 2.2 both satisfy diagonal strict concavity (with a nontrivial choice of weights for the latter), as do atomic splittable congestion games in networks with parallel links (Orda et al., 1993; Sorin and Wan, 2016), multi-user covariance matrix optimization problems in multiple-input and multiple-output (MIMO) systems (Mertikopoulos et al., 2017), and many other problems where online decision-making is the norm. Namely, the class of monotone games contains all strictly convex-concave zero-sum games and all games that admit a (strictly) concave *potential*, i.e., a function $f \colon \mathcal{X} \to \mathbb{R}$ such that $v_i(x) = \nabla_i f(x)$ for all $x \in \mathcal{X}, i \in \mathcal{N}$. In view of all this (and unless explicitly stated otherwise), we will focus throughout on monotone games; for completeness, we also include in the supplement a straightforward second-order test for monotonicity.

## 3    Regularized no-regret learning

We now turn to the learning methods that players could employ to increase their individual rewards in an online manner. Building on Zinkevich's (2003) online gradient descent policy, the most widely used algorithmic schemes for no-regret learning in the context of online convex optimization invariably revolve around the idea of *regularization*. To name but the most well-known paradigms, "following the regularized leader" (FTRL) explicitly relies on best-responding to a regularized aggregate of the reward functions revealed up to a given stage, while online mirror descent (OMD) and its variants use a linear surrogate thereof. All these no-regret policies fall under the general umbrella of "regularized learning" and their origins can be traced back to the seminal *mirror descent* (MD) algorithm of Nemirovski and Yudin (1983).[2]

The basic idea of mirror descent is to generate a new feasible point $x^+$ by taking a so-called "mirror step" from a starting point $x$ along the direction of an "approximate gradient" vector $y$ (which we treat here as an element of the dual space $\mathcal{Y} \equiv \prod_i \mathcal{Y}_i$ of $\mathcal{V} \equiv \prod_i \mathcal{V}_i$).[3] To do so, let $h_i \colon \mathcal{X}_i \to \mathbb{R}$ be a continuous and $K_i$-strongly convex *distance-generating* (or *regularizer*) function, i.e.,

$$h_i(t x_i + (1-t) x_i') \leq t h_i(x_i) + (1-t) h_i(x_i') - \tfrac{1}{2} K_i t(1-t) \|x_i' - x_i\|^2, \tag{3.1}$$

for all $x_i, x_i' \in \mathcal{X}_i$ and all $t \in [0, 1]$. In terms of smoothness (and in a slight abuse of notation) we also assume that the subdifferential of $h_i$ admits a *continuous selection*, i.e., a continuous function $\nabla h_i \colon \operatorname{dom} \partial h_i \to \mathcal{Y}_i$ such that $\nabla h_i(x_i) \in \partial h_i(x_i)$ for all $x_i \in \operatorname{dom} \partial h_i$.[4] Then, letting

$h(x) = \sum_i h_i(x_i)$ for $x \in \mathcal{X}$ (so $h$ is strongly convex with modulus $K = \min_i K_i$), we get a *pseudo-distance* on $\mathcal{X}$ via the relation

$$D(p, x) = h(p) - h(x) - \langle \nabla h(x), p - x \rangle, \tag{3.2}$$

for all $p \in \mathcal{X}$, $x \in \operatorname{dom} \partial h$.

This pseudo-distance is known as the *Bregman divergence* and we have $D(p, x) \geq 0$ with equality if and only if $x = p$; on the other hand, $D$ may fail to be symmetric and/or satisfy the triangle inequality so, in general, it is not a bona fide distance function on $\mathcal{X}$. Nevertheless, we also have $D(p, x) \geq \frac{1}{2} K \|x - p\|^2$ (see the paper's supplement), so the convergence of a sequence $X_n$ to $p$ can be checked by showing that $D(p, X_n) \to 0$. For technical reasons, it will be convenient to also assume the converse, i.e., that $D(p, X_n) \to 0$ when $X_n \to p$. This condition is known in the literature as "Bregman reciprocity" (Chen and Teboulle, 1993), and it will be our blanket assumption in what follows (note that it is trivially satisfied by Examples 3.1 and 3.2 below).

Now, as with true Euclidean distances, $D(p, x)$ induces a *prox-mapping* given by

$$P_x(y) = \underset{x' \in \mathcal{X}}{\arg \min} \{ \langle y, x - x' \rangle + D(x', x) \} \tag{3.3}$$

for all $x \in \operatorname{dom} \partial h$ and all $y \in \mathcal{Y}$. Just like its Euclidean counterpart below, the prox-mapping (3.3) starts with a point $x \in \operatorname{dom} \partial h$ and steps along the dual (gradient-like) vector $y \in \mathcal{Y}$ to produce a new feasible point $x^+ = P_x(y)$. Standard examples of this process are:

**Example 3.1** (Euclidean projections). Let $h(x) = \frac{1}{2} \|x\|_2^2$ denote the Euclidean squared norm. Then, the induced prox-mapping is

$$P_x(y) = \Pi(x + y), \tag{3.4}$$

with $\Pi(x) = \arg \min_{x' \in \mathcal{X}} \|x' - x\|^2$ denoting the standard Euclidean projection onto $\mathcal{X}$. Hence, the update rule $x^+ = P_x(y)$ boils down to a "vanilla", Euclidean projection step along $y$.

**Example 3.2** (Entropic regularization and multiplicative weights). Suppressing the player index for simplicity, let $\mathcal{X}$ be a $d$-dimensional simplex and consider the entropic regularizer $h(x) = \sum_{j=1}^d x_j \log x_j$. The induced pseudo-distance is the so-called *Kullback–Leibler* (KL) divergence $D_{\mathrm{KL}}(p, x) = \sum_{j=1}^d p_j \log(p_j / x_j)$, which gives rise to the prox-mapping

$$P_x(y) = \frac{(x_j \exp(y_j))_{j=1}^d}{\sum_{j=1}^d x_j \exp(y_j)} \tag{3.5}$$

for all $x \in \mathcal{X}^\circ$, $y \in \mathcal{Y}$. The update rule $x^+ = P_x(y)$ is widely known as the *multiplicative weights* (MW) algorithm and plays a central role for learning in multi-armed bandit problems and finite games (Arora et al., 2012; Auer et al., 1995; Freund and Schapire, 1999).

With all this in hand, the multi-agent *mirror descent* (MD) algorithm is given by the recursion

$$X_{n+1} = P_{X_n}(\gamma_n \hat{v}_n), \tag{MD}$$

where $\gamma_n$ is a variable step-size sequence and $\hat{v}_n = (\hat{v}_{i,n})_{i \in \mathcal{N}}$ is a generic feedback sequence of estimated gradients. In the next section, we detail how this sequence is generated with first- or zeroth-order (bandit) feedback.

## 4 First-order vs. bandit feedback

### 4.1 First-order feedback.

A common assumption in the literature is that players are able to obtain gradient information by querying a *first-order oracle* (Nesterov, 2004). i.e., a "black-box" feedback mechanism that outputs an estimate $\hat{v}_i$ of the individual payoff gradient $v_i(x)$ of the $i$-th player at the current action profile $x = (x_i; x_{-i}) \in \mathcal{X}$. This estimate could be either *perfect*, giving $\hat{v}_i = v_i(x)$ for all $i \in \mathcal{N}$, or *imperfect*, returning noisy information of the form $\hat{v}_i = v_i(x) + U_i$ where $U_i$ denotes the oracle's error (random, systematic, or otherwise).

Having access to a perfect oracle is usually a tall order, either because payoff gradients are difficult to compute directly (especially without global knowledge), because they involve an expectation over a possibly unknown probability law, or for any other number of reasons. It is therefore more common to assume that each player has access to a *stochastic oracle* which, when called against a sequence of actions $X_n \in \mathcal{X}$, produces a sequence of gradient estimates $\hat{v}_n = (v_{i,n})_{i \in \mathcal{N}}$ that satisfies the following statistical assumptions:

a) *Unbiasedness:* $\qquad \mathbb{E}[\hat{v}_n \mid \mathcal{F}_n] = v(X_n).$

b) *Finite mean square:* $\quad \mathbb{E}[\|\hat{v}_n\|_*^2 \mid \mathcal{F}_n] \leq V^2$ for some finite $V \geq 0.$

(4.1)

In terms of measurability, the expectation in (4.1) is conditioned on the history $\mathcal{F}_n$ of $X_n$ up to stage $n$; in particular, since $\hat{v}_n$ is generated randomly from $X_n$, it is not $\mathcal{F}_n$-measurable (and hence not adapted). To make this more transparent, we will write $\hat{v}_n = v(X_n) + U_{n+1}$ where $U_n$ is an adapted martingale difference sequence with $\mathbb{E}[\|U_{n+1}\|_*^2 \mid \mathcal{F}_n] \leq \sigma^2$ for some finite $\sigma \geq 0$.

## 4.2 Bandit feedback.

Now, if players don't have access to a first-order oracle – the so-called *bandit* or *payoff-based* framework – they will need to derive an individual gradient estimate from the only information at their disposal: the actual payoffs they receive at each stage. When a function can be queried at multiple points (as few as two in practice), there are efficient ways to estimate its gradient via directional sampling techniques as in Agarwal et al. (2010). In a game-theoretic setting however, multiple-point estimation techniques do not apply because, in general, a player's payoff function depends on the actions of *all* players. Thus, when a player attempts to get a second query of their payoff function, this function may have already changed due to the query of another player – i.e., instead of sampling $u_i(\cdot; x_{-i})$, the $i$-th player would be sampling $u_i(\cdot; x'_{-i})$ for some $x'_{-i} \neq x_{-i}$.

Following Spall (1997) and Flaxman et al. (2005), we posit instead that players rely on a simultaneous perturbation stochastic approximation (SPSA) approach that allows them to estimate their individual payoff gradients $v_i$ based off a *single* function evaluation. In detail, the key steps of this one-shot estimation process for each player $i \in \mathcal{N}$ are:

0. Fix a *query radius* $\delta > 0$.[5]

1. Pick a *pivot point* $x_i \in \mathcal{X}_i$ where player $i$ seeks to estimate their payoff gradient.

2. Draw a vector $z_i$ from the unit sphere $\mathbb{S}_i \equiv \mathbb{S}^{d_i}$ of $\mathcal{V}_i \equiv \mathbb{R}^{d_i}$ and play $\hat{x}_i = x_i + \delta z_i$.[6]

3. Receive $\hat{u}_i = u_i(\hat{x}_i; \hat{x}_{-i})$ and set

$$\hat{v}_i = \frac{d_i}{\delta} \hat{u}_i z_i. \qquad (4.2)$$

By adapting a standard argument based on Stokes' theorem (detailed in the supplement), it can be shown that $\hat{v}_i$ is an unbiased estimator of the individual gradient of the $\delta$-smoothed payoff function

$$u_i^\delta(x) = \frac{1}{\text{vol}(\delta\mathbb{B}_i) \prod_{j \neq i} \text{vol}(\delta\mathbb{S}_j)} \int_{\delta\mathbb{B}_i} \int_{\prod_{j \neq i} \delta\mathbb{S}_j} u_i(x_i + w_i; x_{-i} + z_{-i}) \, dz_1 \cdots dw_i \cdots dz_N \quad (4.3)$$

with $\mathbb{B}_i \equiv \mathbb{B}^{d_i}$ denoting the unit ball of $\mathcal{V}_i$. The Lipschitz continuity of $v_i$ guarantees that $\|\nabla_i u_i - \nabla_i u_i^\delta\|_\infty = \mathcal{O}(\delta)$, so this estimate becomes more and more accurate as $\delta \to 0^+$. On the other hand, the second moment of $\hat{v}_i$ grows as $\mathcal{O}(1/\delta^2)$, implying in turn that the variability of $\hat{v}_i$ grows unbounded as $\delta \to 0^+$. This manifestation of the bias-variance dilemma plays a crucial role in designing no-regret policies with bandit feedback (Flaxman et al., 2005; Kleinberg, 2004), so $\delta$ must be chosen with care.

Before dealing with this choice though, it is important to highlight two feasibility issues that arise with the single-shot SPSA estimate (4.2). The first has to do with the fact that the perturbation direction $z_i$ is chosen from the unit sphere $\mathbb{S}_i$ so it may fail to be tangent to $\mathcal{X}_i$, even when $x_i$ is interior. To iron out this wrinkle, it suffices to sample $z_i$ from the intersection of $\mathbb{S}_i$ with the affine

hull of $\mathcal{X}_i$ in $\mathcal{V}_i$; on that account (and without loss of generality), we will simply assume in what follows that each $\mathcal{X}_i$ is a *convex body* of $\mathcal{V}_i$, i.e., it has nonempty topological interior.

The second feasibility issue concerns the size of the perturbation step: even if $z_i$ is a feasible direction of motion, the query point $\hat{x}_i = x_i + \delta z_i$ may be unfeasible if $x_i$ is too close to the boundary of $\mathcal{X}_i$. For this reason, we will introduce a "safety net" in the spirit of Agarwal et al. (2010), and we will constrain the set of possible pivot points $x_i$ to lie within a suitably shrunk zone of $\mathcal{X}$.

In detail, let $\mathbb{B}_{r_i}(p_i)$ be an $r_i$-ball centered at $p_i \in \mathcal{X}_i$ so that $\mathbb{B}_{r_i}(p_i) \subseteq \mathcal{X}_i$. Then, instead of perturbing $x_i$ by $z_i$, we consider the *feasibility adjustment*

$$w_i = z_i - r_i^{-1}(x_i - p_i), \tag{4.4}$$

and each player plays $\hat{x}_i = x_i + \delta w_i$ instead of $x_i + \delta z_i$. In other words, this adjustment moves each pivot to $x_i^\delta = x_i - r_i^{-1}\delta(x_i - p_i)$, i.e., $\mathcal{O}(\delta)$-closer to the interior base point $p_i$, and then perturbs $x_i^\delta$ by $\delta z_i$. Feasibility of the query point is then ensured by noting that

$$\hat{x}_i = x_i^\delta + \delta z_i = (1 - r_i^{-1}\delta)x_i + r_i^{-1}\delta(p_i + r_i z_i), \tag{4.5}$$

so $\hat{x}_i \in \mathcal{X}_i$ if $\delta/r_i < 1$ (since $p_i + r_i z_i \in \mathbb{B}_{r_i}(p_i) \subseteq \mathcal{X}_i$).

The difference between this estimator and the oracle framework we discussed above is twofold. First, each player's *realized* action is $\hat{x}_i = x_i + \delta w_i$, not $x_i$, so there is a disparity between the point at which payoffs are queried and the action profile where the oracle is called. Second, the resulting estimator $\hat{v}$ is not unbiased, so the statistical assumptions (4.1) for a stochastic oracle do not hold. In particular, given the feasibility adjustment (4.4), the estimate (4.2) with $\hat{x}$ given by (4.5) satisfies

$$\mathbb{E}[\hat{v}_i] = \nabla_i u_i^\delta(x_i^\delta; x_{-i}^\delta), \tag{4.6}$$

so there are *two* sources of systematic error: an $\mathcal{O}(\delta)$ perturbation in the function, and an $\mathcal{O}(\delta)$ perturbation of each player's pivot point from $x_i$ to $x_i^\delta$. Hence, to capture both sources of bias and separate them from the random noise, we will write

$$\hat{v}_i = v_i(x) + U_i + b_i \tag{4.7}$$

where $U_i = \hat{v}_i - \mathbb{E}[\hat{v}_i]$ and $b_i = \nabla_i u_i^\delta(x^\delta) - \nabla_i u_i(x)$. We are thus led to the following manifestation of the bias-variance dilemma: the bias term $b$ in (4.7) is $\mathcal{O}(\delta)$, but the second moment of the noise term $U$ is $\mathcal{O}(1/\delta^2)$; as such, an increase in accuracy (small bias) would result in a commensurate loss of precision (large noise variance). Balancing these two factors will be a key component of our analysis in the next section.

## 5   Convergence analysis and results

Combining the learning framework of Section 3 with the single-shot gradient estimation machinery of Section 4, we obtain the following variant of (MD) with payoff-based, *bandit feedback:*

$$\begin{aligned} \hat{X}_n &= X_n + \delta_n W_n, \\ X_{n+1} &= P_{X_n}(\gamma_n \hat{v}_n). \end{aligned} \tag{MD-b}$$

In the above, the perturbations $W_n$ and the estimates $\hat{v}_n$ are given respectively by (4.4) and (4.2), i.e.,

$$W_{i,n} = Z_{i,n} - r_i^{-1}(X_{i,n} - p_i) \qquad \hat{v}_{i,n} = (d_i/\delta_n)u_i(\hat{X}_n)\,Z_{i,n} \tag{5.1}$$

and $Z_{i,n}$ is drawn independently and uniformly across players at each stage $n$ (see also Algorithm 1 for a pseudocode implementation and Fig. 1 for a schematic representation).

In the rest of this paper, our goal will be to determine the equilibrium convergence properties of this scheme in concave $N$-person games. Our first asymptotic result below shows that, under (MD-b), the players' learning process converges to Nash equilibrium in monotone games:

**Theorem 5.1.** *Suppose that the players of a monotone game $\mathcal{G} \equiv \mathcal{G}(\mathcal{N}, \mathcal{X}, u)$ follow* (MD-b) *with step-size $\gamma_n$ and query radius $\delta_n$ such that*

$$\lim_{n \to \infty} \gamma_n = \lim_{n \to \infty} \delta_n = 0, \quad \sum_{n=1}^{\infty} \gamma_n = \infty, \quad \sum_{n=1}^{\infty} \gamma_n \delta_n < \infty, \quad and \quad \sum_{n=1}^{\infty} \frac{\gamma_n^2}{\delta_n^2} < \infty. \tag{5.2}$$

*Then, the sequence of realized actions $\hat{X}_n$ converges to Nash equilibrium with probability 1.*

---

**Algorithm 1:** Multi-agent mirror descent with bandit feedback  (player indices suppressed)

---

**Require:** step-size $\gamma_n > 0$, query radius $\delta_n > 0$, safety ball $\mathbb{B}_r(p) \subseteq \mathcal{X}$
1: choose $X \in \operatorname{dom}\partial h$                                        # initialization
2: **repeat** at each stage $n = 1, 2, \ldots$
3:     draw $Z$ uniformly from $\mathbb{S}^d$                              # perturbation direction
4:     set $W \leftarrow Z - r^{-1}(X - p)$                                   # query direction
5:     play $\hat{X} \leftarrow X + \delta_n W$                                   # choose action
6:     receive $\hat{u} \leftarrow u(\hat{X})$                                      # get payoff
7:     set $\hat{v} \leftarrow (d/\delta_n)\hat{u} \cdot Z$                            # estimate gradient
8:     update $X \leftarrow P_X(\gamma_n \hat{v})$                               # update pivot
9: **until** end

---

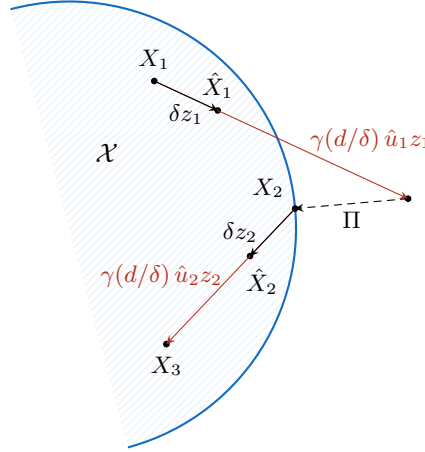

**Figure 1:** Schematic representation of Algorithm 1 with ordinary, Euclidean projections. To reduce visual clutter, we did not include the feasibility adjustment $r^{-1}(x - p)$ in the action selection step $X_n \mapsto \hat{X}_n$.

Even though the setting is different, the conditions (5.2) for the tuning of the algorithm's parameters are akin to those encountered in Kiefer–Wolfowitz stochastic approximation schemes and serve a similar purpose. First, the conditions $\lim_{n\to\infty} \gamma_n = 0$ and $\sum_{n=1}^{\infty} \gamma_n = \infty$ respectively mitigate the method's inherent randomness and ensure a horizon of sufficient length. The requirement $\lim_{n\to\infty} \delta_n = 0$ is also straightforward to explain: as players accrue more information, they need to decrease the sampling bias in order to have any hope of converging. However, as we discussed in Section 4, decreasing $\delta$ also increases the variance of the players' gradient estimates, which might grow to infinity as $\delta \to 0$. The crucial observation here is that new gradients enter the algorithm with a weight of $\gamma_n$ so the aggregate bias after $n$ stages is of the order of $\mathcal{O}(\sum_{k=1}^{n} \gamma_k \delta_k)$ and its variance is $\mathcal{O}(\sum_{k=1}^{n} \gamma_k^2/\delta_k^2)$. If these error terms can be controlled, there is an underlying drift that emerges over time and which steers the process to equilibrium. We make this precise in the supplement by using a suitably adjusted variant of the Bregman divergence as a quasi-Féjér energy function for (MD-b) and relying on a series of (sub)martingale convergence arguments to establish the convergence of $\hat{X}_n$ (first as a subsequence, then with probability 1).

Of course, since Theorem 5.1 is asymptotic in nature, it is not clear how to choose $\gamma_n$ and $\delta_n$ so as to optimize the method's convergence rate. Heuristically, if we take schedules of the form $\gamma_n = \gamma/n^p$ and $\delta_n = \delta/n^q$ with $\gamma, \delta > 0$ and $0 < p, q \leq 1$, the only conditions imposed by (5.2) are $p + q > 1$ and $p - q > 1/2$. However, as we discussed above, the aggregate bias in the algorithm after $n$ stages is $\mathcal{O}(\sum_{k=1}^{n} \gamma_n \delta_n) = \mathcal{O}(1/n^{p+q-1})$ and its variance is $\mathcal{O}(\sum_{k=1}^{n} \gamma_k^2/\delta_k^2) = \mathcal{O}(1/n^{2p-2q-1})$: if the conditions (5.2) are satisfied, both error terms vanish, but they might do so at very different rates. By equating these exponents in order to bridge this gap, we obtain $q = p/3$; moreover, since the single-shot SPSA estimator (4.2) introduces a $\Theta(\delta_n)$ random perturbation, $q$ should be taken as large as possible to ensure that this perturbation vanishes at the fastest possible rate. As a result, the most suitable choice for $p$ and $q$ seems to be $p = 1$, $q = 1/3$, leading to an error bound of $\mathcal{O}(1/n^{1/3})$.

We show below that this bound is indeed attainable for games that are *strongly monotone*, i.e., they satisfy the following stronger variant of diagonal strict concavity:

$$\sum_{i \in \mathcal{N}} \lambda_i \langle v_i(x') - v_i(x), x'_i - x_i \rangle \leq -\frac{\beta}{2} \|x - x'\|^2 \qquad (\beta\text{-DSC})$$

for some $\lambda_i, \beta > 0$ and for all $x, x' \in \mathcal{X}$. Focusing for expository reasons on the most widely used, Euclidean incarnation of the method (Example 3.1), we have:

**Theorem 5.2.** *Let $x^*$ be the* (*necessarily unique*) *Nash equilibrium of a $\beta$-strongly monotone game. If the players follow* (MD-b) *with Euclidean projections and parameters $\gamma_n = \gamma/n$ and $\delta_n = \delta/n^{1/3}$ with $\gamma > 1/(3\beta)$ and $\delta > 0$, we have*

$$\mathbb{E}[\|\hat{X}_n - x^*\|^2] = \mathcal{O}(n^{-1/3}). \qquad (5.3)$$

Theorem 5.2 is our main finite-time analysis result, so some remarks are in order. First, the step-size schedule $\gamma_n \propto 1/n$ is not required to obtain an $\mathcal{O}(n^{-1/3})$ convergence rate: as we show in the paper's supplement, more general schedules of the form $\gamma_n \propto 1/n^p$ and $\delta_n \propto 1/n^q$ with $p > 3/4$ and $q = p/3 > 1/4$, still guarantee an $\mathcal{O}(n^{-1/3})$ rate of convergence for (MD-b). To put things in perspective, we also show in the supplement that if (MD) is run with first-order oracle feedback satisfying the statistical assumptions (4.1), the rate of convergence becomes $\mathcal{O}(1/n)$. Viewed in this light, the price for not having access to gradient information is no higher than $\mathcal{O}(n^{-2/3})$ in terms of the players' equilibration rate.

Finally, it is also worth comparing the bound (5.3) to the attainable rates for stochastic convex optimization (the single-player case). For problems with objectives that are both strongly convex and smooth, Agarwal et al. (2010) attained an $\mathcal{O}(n^{-1/2})$ convergence rate with bandit feedback, which Shamir (2013) showed is unimprovable. Thus, in the single-player case, the bound (5.3) is off by $n^{1/6}$ and coincides with the bound of Agarwal et al. (2010) for strongly convex functions that are not necessarily smooth. One reason for this gap is that the $\Theta(n^{-1/2})$ bound of Shamir (2013) concerns the smoothed-out time average $\bar{X}_n = n^{-1} \sum_{k=1}^n X_k$, while our analysis concerns the sequence of *realized actions* $\hat{X}_n$. This difference is semantically significant: In optimization, the query sequence is just a means to an end, and only the algorithm's output matters (i.e., $\bar{X}_n$). In a game-theoretic setting however, it is the players' *realized* actions that determine their rewards at each stage, so the figure of merit is the actual sequence of play $\hat{X}_n$. This sequence is more difficult to control, so this disparity is, perhaps, not too surprising; nevertheless, we believe that this gap can be closed by using a more sophisticated single-shot estimate, e.g., as in Ghadimi and Lan (2013). We defer this analysis to the future.

## 6  Concluding remarks

The most sensible choice for agents who are oblivious to the presence of each other (or who are simply conservative), is to deploy a no-regret learning algorithm. With this in mind, we studied the long-run behavior of individual regularized no-regret learning policies and we showed that, in monotone games, play converges to equilibrium with probability 1, and the rate of convergence almost matches the optimal rates of *single-agent*, stochastic convex optimization. Nevertheless, several questions remain open: whether there is an intrinsic information-theoretic obstacle to bridging this gap; whether our convergence rate estimates hold with high probability (and not just in expectation); and whether our analysis extends to a fully decentralized setting where the players' updates need not be synchronous. We intend to address these questions in future work.

## Acknowledgments

M. Bravo gratefully acknowledges the support provided by FONDECYT grant 11151003. P. Mertikopoulos was partially supported by the Huawei HIRP flagship grant ULTRON, and the French National Research Agency (ANR) grant ORACLESS (ANR–16–CE33–0004–01). Part of this work was carried out with financial support by the ECOS project C15E03.

## Footnotes

[1] We adopt here the standard convention of treating $v_i(x)$ as an element of the dual space $\mathcal{Y}_i \equiv \mathcal{V}_i^*$ of $\mathcal{V}_i$, with $\langle y_i, x_i \rangle$ denoting the duality pairing between $y_i \in \mathcal{Y}_i$ and $x_i \in \mathcal{X}_i \subseteq \mathcal{V}_i$.

[2]In a utility maximization setting, mirror descent should be called mirror *ascent* because players seek to *maximize* their rewards (as opposed to *minimizing* their losses). Nonetheless, we keep the term "descent" throughout because, despite the role reversal, it is the standard name associated with the method.

[3]For concreteness (and in a slight abuse of notation), we assume in what follows that $\mathcal{V}$ is equipped with the product norm $\|x\|^2 = \sum_i \|x_i\|^2$ and $\mathcal{Y}$ with the dual norm $\|y\|_* = \max\{\langle y, x \rangle : \|x\| \leq 1\}$.

[4]Recall here that the subdifferential of $h_i$ at $x_i \in \mathcal{X}_i$ is defined as $\partial h_i(x_i) \equiv \{y_i \in \mathcal{Y}_i : h_i(x_i') \geq h_i(x_i) + \langle y_i, x_i' - x_i \rangle$ for all $x_i' \in \mathcal{V}_i\}$, with the standard convention that $h_i(x_i) = +\infty$ if $x_i \in \mathcal{V}_i \setminus \mathcal{X}_i$. By standard results, the domain of subdifferentiability $\partial h_i \equiv \{x_i \in \mathcal{X}_i : \partial h_i \neq \varnothing\}$ of $h_i$ satisfies $\mathcal{X}_i^\circ \subseteq \operatorname{dom} \partial h_i \subseteq \mathcal{X}_i$.

[5]For simplicity, we take $\delta$ equal for all players; the extension to player-specific $\delta$ is straightforward, so we omit it.

[6]We tacitly assume here that the query directions $z_i \in \mathbb{S}^{d_i}$ are drawn independently across players.

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
