[Supplementary Material · Supplement.pdf]

# Supplementary material for
# Bandit Learning in Concave $N$-Person Games

## A  Preamble

For completeness, we briefly reproduce here some basic definitions concerning the most important elements of our paper.

First, given a $K$-strongly convex regularizer $h\colon \mathcal{X} \to \mathbb{R}$ (the player index $i$ is suppressed for simplicity), the associated Bregman divergence is defined as

$$D(p,x) = h(p) - h(x) - \langle \nabla h(x), p - x \rangle \tag{A.1}$$

with $\nabla h(x)$ denoting a continuous selection of $\partial h(x)$. The induced prox-mapping is then given by

$$\begin{aligned} P_x(y) &= \operatorname*{arg\,min}_{x' \in \mathcal{X}} \{\langle y, x - x' \rangle + D(x', x)\} \\ &= \operatorname*{arg\,max}_{x' \in \mathcal{X}} \{\langle y + \nabla h(x), x' \rangle - h(x')\} \end{aligned} \tag{A.2}$$

and is defined for all $x \in \operatorname{dom} \partial h$, $y \in \mathcal{Y}$ (recall here that $\mathcal{Y} \equiv \mathcal{V}^*$ denotes the dual of the ambient vector space $\mathcal{V}$ in which the game's action space $\mathcal{X}$ is embedded).[1]

With all this at hand, the multi-agent mirror descent algorithm with bandit feedback is defined as follows:

$$\begin{aligned} \hat{X}_n &= X_n + \delta_n W_n, \\ X_{n+1} &= P_{X_n}(\gamma_n \hat{v}_n). \end{aligned} \tag{MD-b}$$

where the perturbation $W_n$ and the estimate $\hat{v}_n$ are given respectively by

$$W_{i,n} = Z_{i,n} - r_i^{-1}(X_{i,n} - p_i) \qquad \hat{v}_{i,n} = (d_i/\delta_n) u_i(\hat{X}_n) Z_{i,n}. \tag{A.3}$$

In the above, the query directions $Z_{i,n}$ are drawn independently and uniformly across players at each stage $n$ from the corresponding unit sphere; finally, $\mathbb{B}_{r_i}(p_i)$ denotes a ball that is entirely contained in $\mathcal{X}_i$. For a schematic representation, see also Fig. 1.

## B  Monotone games

We now turn to the game-theoretic examples of Section 2. Before studying them in detail, it will be convenient to introduce a straightforward second-order test for monotonicity based on the game's Hessian matrix.

Specifically, extending the notion of the Hessian of an ordinary (scalar) function, the ($\lambda$-*weighted*) *Hessian* of a game $\mathcal{G}$ is defined as the block matrix $H_{\mathcal{G}}(x; \lambda) = (H_{ij}(x; \lambda))_{i,j \in \mathcal{N}}$ with blocks

$$H_{ij}(x; \lambda) = \frac{\lambda_i}{2} \nabla_j \nabla_i u_i(x) + \frac{\lambda_j}{2} (\nabla_i \nabla_j u_j(x))^\top. \tag{B.1}$$

**Figure 1:** Schematic representation of (MD-b) with ordinary, Euclidean projections. To reduce visual clutter, we did not include the feasibility adjustment $r^{-1}(x - p)$ in the action selection step $X_n \mapsto \hat{X}_n$.

As was shown by Rosen (1965, Theorem 6), $\mathcal{G}$ satisifes (DSC) with weight vector $\lambda$ whenever $z^\top H_\mathcal{G}(x; \lambda) z < 0$ for all $x \in \mathcal{X}$ and all nonzero $z \in \mathcal{V} \equiv \prod_i \mathcal{V}_i$ that are tangent to $\mathcal{X}$ at $x$.[2] It is thus common to check for monotonicity by taking $\lambda_i = 1$ for all $i \in \mathcal{N}$ and verifying whether the unweighted Hessian of $\mathcal{G}$ is negative-definite on the affine hull of $\mathcal{X}$.

**Cournot competition (Example 2.1).** In the standard Cournot oligopoly model described in the main body of the paper, the players' payoff functions are given by

$$u_i(x) = x_i\big(a - b\textstyle\sum_j x_j\big) - c_i x_i. \tag{B.2}$$

Consequently, a simple differentiation yields

$$H_{ij}(x) = \frac{1}{2}\frac{\partial^2 u_i}{\partial x_i \partial x_j} + \frac{1}{2}\frac{\partial^2 u_j}{\partial x_j \partial x_i} = -b(1 + \delta_{ij}), \tag{B.3}$$

where $\delta_{ij} = \mathbb{1}\{i = j\}$ is the Kronecker delta. This matrix is clearly negative-definite, so the game is monotone.

**Resource allocation auctions (Example 2.2).** In our auction-theoretic example, the players' payoff functions are given by

$$u_i(x_i; x_{-i}) = \sum_{s \in \mathcal{S}} \left[ \frac{g_i q_s x_{is}}{c_s + \sum_{j \in \mathcal{N}} x_{js}} - x_{is} \right] \tag{B.4}$$

To prove monotonicity in this example, we will consider the following criterion due to Goodman (1980): a game $\mathcal{G}$ satisfies (DSC) with weights $\lambda_i$, $i \in \mathcal{N}$, if:

    *a)* Each payoff function $u_i$ is strictly concave in $x_i$ and convex in $x_{-i}$.

    *b)* The function $\sum_{i \in \mathcal{N}} \lambda_i u_i(x)$ is concave in $x$.

Since the function $\phi(x) = x/(c + x)$ is strictly concave in $x$ for all $c > 0$, the first condition above is trivial to verify. For the second, letting $\lambda_i = 1/g_i$ gives

$$\sum_{i \in \mathcal{N}} \lambda_i u_i(x) = \sum_{i \in \mathcal{N}} \sum_{s \in \mathcal{S}} \frac{q_s x_{is}}{c_s + \sum_{j \in \mathcal{N}} x_{js}} - \sum_{i \in \mathcal{N}} \sum_{s \in \mathcal{S}} x_{is}$$

$$= \sum_{s \in \mathcal{S}} q_s \frac{\sum_{i \in \mathcal{N}} x_{is}}{c_s + \sum_{i \in \mathcal{N}} x_{is}} - \sum_{i \in \mathcal{N}} \sum_{s \in \mathcal{S}} x_{is}. \tag{B.5}$$

Since the summands above are all concave in their respective arguments, our claim follows.

## C  Properties of Bregman proximal mappings

In this appendix, we provide some auxiliary results and estimates that are used throughout the convergence analysis of Appendix D. Some of the results we present here are not new (see e.g., Nemirovski et al., 2009); however, the set of hypotheses used to obtain them varies widely in the literature, so we provide all proofs for completeness.

In what follows, we will make frequent use of the convex conjugate $h^*\colon \mathcal{Y} \to \mathbb{R}$ of $h$, defined here as

$$h^*(y) = \max_{x \in \mathcal{X}}\{\langle y, x \rangle - h(x)\}. \tag{C.1}$$

By standard results in convex analysis (Rockafellar, 1970, Chap. 26), $h^*$ is differentiable on $\mathcal{Y}$ and its gradient satisfies the identity

$$\nabla h^*(y) = \arg\max_{x \in \mathcal{X}}\{\langle y, x \rangle - h(x)\}. \tag{C.2}$$

For notational convenience, we will also write

$$Q(y) = \nabla h^*(y) \tag{C.3}$$

and we will refer to $Q\colon \mathcal{Y} \to \mathcal{X}$ as the *mirror map* generated by $h$.

Together with the prox-mapping induced by $h$, all these notions are related as follows:

**Lemma 1.** *Let $h$ be a regularizer on $\mathcal{X}$. Then, for all $x \in \operatorname{dom} \partial h$, $y \in \mathcal{Y}$, we have:*

*a)*  $x = Q(y) \iff y \in \partial h(x).$ $\hspace{4cm}$ (C.4a)

*b)*  $x^+ = P_x(y) \iff \nabla h(x) + y \in \partial h(x^+) \iff x^+ = Q(\nabla h(x) + y).$ $\hspace{0.5cm}$ (C.4b)

*Finally, if $x = Q(y)$ and $p \in \mathcal{X}$, we have*

$$\langle \nabla h(x), x - p \rangle \le \langle y, x - p \rangle. \tag{C.5}$$

*Remark.* Note that (C.4b) directly implies that $\partial h(x^+) \ne \varnothing$, i.e., $x^+ \in \operatorname{dom}\partial h$. An immediate consequence of this is that the update rule $x \leftarrow P_x(y)$ is *well-posed*, i.e., it can be iterated in perpetuity.

*Proof of Lemma 1.* To prove (C.4a), note that $x$ solves (C.2) if and only if $y - \partial h(x) \ni 0$, i.e., if and only if $y \in \partial h(x)$. Similarly, for (C.4b), comparing (A.2) and (C.1), we see that $x^+$ solves (A.2) if and only if $\nabla h(x) + y \in \partial h(x^+)$, i.e., if and only if $x^+ = Q(\nabla h(x) + y)$.

For the inequality (C.5), it suffices to show it holds for interior $p \in \mathcal{X}^\circ$ (by continuity). To do so, let

$$\phi(t) = h(x + t(p - x)) - [h(x) + \langle y, x + t(p - x)\rangle]. \tag{C.6}$$

Since $h$ is strongly convex and $y \in \partial h(x)$ by (C.4a), it follows that $\phi(t) \ge 0$ with equality if and only if $t = 0$. Moreover, note that $\psi(t) = \langle \nabla h(x + t(p - x)) - y, p - x\rangle$ is a continuous selection of subgradients of $\phi$. Given that $\phi$ and $\psi$ are both continuous on $[0, 1]$, it follows that $\phi$ is continuously differentiable and $\phi' = \psi$ on $[0, 1]$. Thus, with $\phi$ convex and $\phi(t) \ge 0 = \phi(0)$ for all $t \in [0, 1]$, we conclude that $\phi'(0) = \langle \nabla h(x) - y, p - x\rangle \ge 0$, from which our claim follows. $\hspace{1cm} \square$

We continue with some basic relations connecting the Bregman divergence relative to a target point before and after a prox step. The basic ingredient for this is a generalization of the law of cosines which is known in the literature as the "three-point identity" (Chen and Teboulle, 1993):

**Lemma 2.** *Let $h$ be a regularizer on $\mathcal{X}$. Then, for all $p \in \mathcal{X}$ and all $x, x' \in \operatorname{dom}\partial h$, we have*

$$D(p, x') = D(p, x) + D(x, x') + \langle \nabla h(x') - \nabla h(x), x - p\rangle. \tag{C.7}$$

*Proof.* By definition, we get:

$$\begin{aligned}
D(p, x') &= h(p) - h(x') - \langle \nabla h(x'), p - x'\rangle \\
D(p, x) &= h(p) - h(x) - \langle \nabla h(x), p - x\rangle \\
D(x, x') &= h(x) - h(x') - \langle \nabla h(x'), x - x'\rangle.
\end{aligned} \tag{C.8}$$

The lemma then follows by adding the two last lines and subtracting the first. $\hspace{1cm} \square$

70    With all this at hand, we have the following upper and lower bounds:

71    **Proposition 3.** *Let $h$ be a $K$-strongly convex regularizer on $\mathcal{X}$, fix some $p \in \mathcal{X}$, and let $x^+ = P_x(y)$*
72    *for $x \in \operatorname{dom} \partial h$, $y \in \mathcal{Y}$. Then, we have:*

$$D(p, x) \geq \frac{K}{2}\|x - p\|^2. \tag{C.9a}$$

$$D(p, x^+) \leq D(p, x) - D(x^+, x) + \langle y, x^+ - p \rangle \tag{C.9b}$$

$$\leq D(p, x) + \langle y, x - p \rangle + \frac{1}{2K}\|y\|_*^2 \tag{C.9c}$$

73    *Proof of* (C.9a). By the strong convexity of $h$, we get

$$h(p) \geq h(x) + \langle \nabla h(x), p - x \rangle + \frac{K}{2}\|p - x\|^2 \tag{C.10}$$

74    so (C.9a) follows by gathering all terms involving $h$ and recalling the definition of $D(p, x)$.    $\square$

75    *Proof of* (C.9b) *and* (C.9c). By the three-point identity (C.7), we readily obtain

$$D(p, x) = D(p, x^+) + D(x^+, x) + \langle \nabla h(x) - \nabla h(x^+), x^+ - p \rangle, \tag{C.11}$$

76    and hence:

$$D(p, x^+) = D(p, x) - D(x^+, x) + \langle \nabla h(x^+) - \nabla h(x), x^+ - p \rangle$$
$$\leq D(p, x) - D(x^+, x) + \langle y, x^+ - p \rangle, \tag{C.12}$$

77    where, in the last step, we used (C.5) and the fact that $x^+ = Q(\nabla h(x) + y)$, by (C.4b), since
78    $x^+ = P_x(y)$. The above is just (C.9b), so the first part of our proof is complete.

79    To proceed with the proof of (C.9c), note that (C.12) gives

$$D(p, x^+) \leq D(p, x) + \langle y, x - p \rangle + \langle y, x^+ - x \rangle - D(x^+, x). \tag{C.13}$$

80    By Young's inequality (Rockafellar, 1970), we also have

$$\langle y, x^+ - x \rangle \leq \frac{K}{2}\|x^+ - x\|^2 + \frac{1}{2K}\|y\|_*^2, \tag{C.14}$$

81    and hence

$$D(p, x^+) \leq D(p, x) + \langle y, x - p \rangle + \frac{1}{2K}\|y\|_*^2 + \frac{K}{2}\|x^+ - x\|^2 - D(x^+, x)$$
$$\leq D(p, x) + \langle y, x - p \rangle + \frac{1}{2K}\|y\|_*^2, \tag{C.15}$$

82    with the last step following from Lemma 1 after plugging in $x$ in place of $p$.    $\square$

## 83   D   Asymptotic convergence analysis

84    Our goal in this appendix is to prove Theorem 5.1. Since this is our basic asymptotic convergence
85    result, we reproduce it below for convenience:

86    **Theorem.** *Suppose that the players of a monotone game $\mathcal{G} \equiv \mathcal{G}(\mathcal{N}, \mathcal{X}, u)$ follow* (MD-b) *with*
87    *step-size $\gamma_n$ and query radius $\delta_n$ such that*

$$\lim_{n \to \infty} \gamma_n = \lim_{n \to \infty} \delta_n = 0, \quad \sum_{n=1}^{\infty} \gamma_n = \infty, \quad \sum_{n=1}^{\infty} \gamma_n \delta_n < \infty, \quad and \quad \sum_{n=1}^{\infty} \frac{\gamma_n^2}{\delta_n^2} < \infty. \tag{D.1}$$

88    *Then, the sequence of realized actions $\hat{X}_n$ converges to Nash equilibrium with probability* 1.

Our proof strategy will be based on a two-pronged approach. First, we will show that the pivot sequence $X_n$ satisfies a "quasi-Fejér" property (Combettes, 2001; Combettes and Pesquet, 2015) with respect to the Bregman divergence. This quasi-Fejér property allows us to show that the Bregman divergence $D(x^*, X_n)$ with respect to a Nash equilibrium $x^*$ of $\mathcal{G}$ converges. To show that this limit is actually zero for *some* Nash equilibrium, we prove that, with probability 1, the sequence $X_n$ admits a (random) subsequence that converges to a Nash equilibrium. The theorem then follows by combining these two results.

To carry all this out, we begin with an auxiliary lemma for the simultaneous perturbation stochastic approximation (SPSA) estimation process of Section 4:

**Lemma 4.** *The SPSA estimator* $\hat{v} = (\hat{v}_i)_{i \in \mathcal{N}}$ *given by* (4.2) *satisfies*

$$\mathbb{E}[\hat{v}_i] = \nabla_i\, u_i^\delta, \tag{D.2}$$

*with* $u_i^\delta$ *as in* (4.3). *Moreover, we have* $\|\nabla_i\, u_i^\delta - \nabla_i\, u_i\|_\infty = \mathcal{O}(\delta)$.

*Proof.* By the independence of the sampling directions $z_i$, $i \in \mathcal{N}$, we have

$$
\begin{aligned}
\mathbb{E}[\hat{v}_i] &= \frac{d_i/\delta}{\prod_j \mathrm{vol}(\mathbb{S}_j)} \int_{\mathbb{S}_1} \cdots \int_{\mathbb{S}_N} u_i(x_1 + \delta z_1, \ldots, x_N + \delta z_N) z_i \; dz_1 \cdots dz_N \\
&= \frac{d_i/\delta}{\prod_j \mathrm{vol}(\delta\mathbb{S}_j)} \int_{\delta\mathbb{S}_1} \cdots \int_{\delta\mathbb{S}_N} u_i(x_1 + z_1, \ldots, x_N + z_N) \frac{z_i}{\|z_i\|} \; dz_1 \cdots dz_N \\
&= \frac{d_i/\delta}{\prod_j \mathrm{vol}(\delta\mathbb{S}_j)} \int_{\delta\mathbb{S}_i} \int_{\prod_{j \neq i} \delta\mathbb{S}_j} u_i(x_i + z_i; x_{-i} + z_{-i}) \frac{z_i}{\|z_i\|} \; dz_i \, dz_{-i} \\
&= \frac{d_i/\delta}{\prod_j \mathrm{vol}(\delta\mathbb{S}_j)} \int_{\delta\mathbb{B}_i} \int_{\prod_{j \neq i} \delta\mathbb{S}_j} \nabla_i\, u_i(x_i + w_i; x_{-i} + z_{-i}) \; dw_i \, dz_{-i},
\end{aligned}
\tag{D.3}
$$

where, in the last line, we used the identity

$$\nabla \int_{\delta\mathbb{B}} f(x + w)\, dw = \int_{\delta\mathbb{S}} f(x + z) \frac{z}{\|z\|} \; dz \tag{D.4}$$

which, in turn, follows from Stokes' theorem (Flaxman et al., 2005; Lee, 2003). Since $\mathrm{vol}(\delta\mathbb{B}_i) = (\delta/d_i)\,\mathrm{vol}(\delta\mathbb{S}_i)$, the above yields $\mathbb{E}[\hat{v}_i] = \nabla_i\, u_i^\delta$ with $u_i^\delta$ given by (4.3).

For the second part of the lemma, let $L_i$ denote the Lipschitz constant of $v_i$, i.e., $\|v_i(x') - v_i(x)\|_* \leq L_i \|x' - x\|$ for all $x, x' \in \mathcal{X}$. Then, for all $w_i \in \delta\mathbb{B}_i$ and all $z_j \in \delta\mathbb{S}_j$, $j \neq i$, we have

$$\|\nabla_i\, u_i(x_i + w_i; x_{-i} + z_{-i}) - \nabla_i\, u_i(x)\| \leq L_i \sqrt{\|w_i\|^2 + \sum\nolimits_{j \neq i} \|z_j\|^2} \leq L_i \sqrt{N} \delta. \tag{D.5}$$

Our assertion then follows by integrating and differentiating under the integral sign. $\qquad\square$

With this basic estimate at hand, we proceed to establish the convergence of the Bregman divergence relative to the game's Nash equilibria:

**Proposition 5.** *Let* $x^*$ *be a Nash equilibrium of* $\mathcal{G}$. *Then, with assumptions as in Theorem 5.1, the Bregman divergence* $D(x^*, X_n)$ *converges (a.s.) to a finite random variable* $D_\infty$.

*Remark.* For expository reasons, we tacitly assume above (and in what follows) that $\mathcal{G}$ satisfies (DSC) with weights $\lambda_i = 1$ for all $i \in \mathcal{N}$. If this is not the case, the Bregman divergence $D(p, x)$ should be replaced by the weight-adjusted variant

$$D^\lambda(p, x) = \sum_{i \in \mathcal{N}} \lambda_i D(p_i, x_i). \tag{D.6}$$

Since this adjustment would force us to carry around all player indices, the presentation would become significantly more cumbersome; to avoid this, we stick with the simpler, unweighted case.

*Proof.* Let $D_n = D(x^*, X_n)$ for some Nash equilibrium $x^*$ of $\mathcal{G}$ and write

$$\hat{v}_n = v(X_n) + U_{n+1} + b_n, \tag{D.7}$$

where, recalling the setup of Section 4 in the main body of the paper, the noise process $U_{n+1} = \hat{v}_n - \mathbb{E}[\hat{v}_n \,|\, \mathcal{F}_n]$ is an $\mathcal{F}_n$-adapted martingale difference sequence and $b_n = v^{\delta_n}(X_n^{\delta_n}) - v(X_n)$ denotes the systematic bias of the estimator $\hat{v}_n$.[3] Then, by Proposition 3, we have

$$
\begin{aligned}
D_{n+1} = D(x^*, P_{X_n}(\gamma_n \hat{v}_n)) &\leq D(x^*, X_n) + \gamma_n \langle \hat{v}_n, X_n - x^* \rangle + \frac{\gamma_n^2}{2K} \|\hat{v}_n\|_*^2 \\
&= D_n + \gamma_n \langle v(X_n) + U_{n+1} + b_n, X_n - x^* \rangle + \frac{\gamma_n^2}{2K} \|\hat{v}_n\|_*^2 \\
&\leq D_n + \gamma_n \xi_{n+1} + \gamma_n r_n + \frac{\gamma_n^2}{2K} \|\hat{v}_n\|_*^2,
\end{aligned}
\tag{D.8}
$$

where, in the last line, we set $\xi_{n+1} = \langle U_{n+1}, X_n - x^* \rangle$, $r_n = \langle b_n, X_n - x^* \rangle$, and we used the variational characterization (VI) of Nash equilibria of monotone games. Thus, conditioning on $\mathcal{F}_n$ and taking expectations, we get

$$
\begin{aligned}
\mathbb{E}[D_{n+1} \,|\, \mathcal{F}_n] &\leq D_n + \mathbb{E}[\xi_{n+1} \,|\, \mathcal{F}_n] + \gamma_n \mathbb{E}[r_n \,|\, \mathcal{F}_n] + \frac{\gamma_n^2}{2K} \mathbb{E}[\|\hat{v}_n\|_*^2 \,|\, \mathcal{F}_n] \\
&\leq D_n + \gamma_n \mathbb{E}[r_n \,|\, \mathcal{F}_n] + \frac{V^2}{2K} \frac{\gamma_n^2}{\delta_n^2}.
\end{aligned}
\tag{D.9}
$$

where we set $V^2 = \sum_i d_i^2 \max_{x \in \mathcal{X}} |u_i(x)|^2$ and we used the fact that $X_n$ is $\mathcal{F}_n$-measurable, so

$$
\mathbb{E}[\xi_{n+1} \,|\, \mathcal{F}_n] = \langle \mathbb{E}[U_{n+1} \,|\, \mathcal{F}_n], X_n - x^* \rangle = 0.
\tag{D.10}
$$

Finally, by Lemma 4, we have

$$
\|b_n\|_* = \|v^{\delta_n}(X_n^{\delta_n}) - v(X_n)\|_* \leq \|v^{\delta_n}(X_n^{\delta_n}) - v(X_n^{\delta_n})\|_* + \|v(X_n^{\delta_n}) - v(X_n)\|_* = \mathcal{O}(\delta_n),
\tag{D.11}
$$

where we used the fact that $v$ is Lipschitz continuous and $\|v^\delta - v\|_\infty = \mathcal{O}(\delta)$. This shows that there exists some $B > 0$ such that $r_n \leq B\delta_n$; as a consequence, we obtain

$$
\mathbb{E}[D_{n+1} \,|\, \mathcal{F}_n] \leq D_n + B\gamma_n \delta_n + \frac{V^2}{2K} \frac{\gamma_n^2}{\delta_n^2}.
\tag{D.12}
$$

Now, letting $R_n = D_n + \sum_{k=n}^{\infty}[B\gamma_k \delta_k + (2K)^{-1} V^2 \gamma_k^2/\delta_k^2]$, the estimate (D.8) gives

$$
\begin{aligned}
\mathbb{E}[R_{n+1} \,|\, \mathcal{F}_n] &= \mathbb{E}[D_{n+1} \,|\, \mathcal{F}_n] + \sum_{k=n+1}^{\infty} \left[ B\gamma_k \delta_k + \frac{V^2}{2K} \frac{\gamma_k^2}{\delta_k^2} \right] \\
&\leq D_n + B\gamma_n \delta_n + \frac{V^2}{2K} \frac{\gamma_n^2}{\delta_n^2} + \sum_{k=n+1}^{\infty} \left[ B\gamma_k \delta_k + \frac{V^2}{2K} \frac{\gamma_k^2}{\delta_k^2} \right] \\
&\leq D_n + \sum_{k=n}^{\infty} \left[ B\gamma_k \delta_k + \frac{V^2}{2K} \frac{\gamma_k^2}{\delta_k^2} \right] \\
&= R_n,
\end{aligned}
\tag{D.13}
$$

i.e., $R_n$ is an $\mathcal{F}_n$-adapted supermartingale.[4] Since the series $\sum_{n=1}^{\infty} \gamma_n \delta_n$ and $\sum_{n=1}^{\infty} \gamma_n^2/\delta_n^2$ are both summable, it follows that

$$
\mathbb{E}[R_n] = \mathbb{E}[\mathbb{E}[R_n \,|\, \mathcal{F}_{n-1}]] \leq \mathbb{E}[R_{n-1}] \leq \cdots \leq \mathbb{E}[R_1] \leq \mathbb{E}[D_1] + \sum_{n=1}^{\infty} \left[ B\gamma_n \delta_n + \frac{V^2}{2K} \frac{\gamma_n^2}{\delta_n^2} \right] < \infty
\tag{D.14}
$$

i.e., $R_n$ is uniformly bounded in $L^1$. Thus, by Doob's convergence theorem for supermartingales (Hall and Heyde, 1980, Theorem 2.5), it follows that $R_n$ converges (a.s.) to some finite random variable $R_\infty$. In turn, by inverting the definition of $R_n$, it follows that $D_n$ converges (a.s.) to some random variable $D_\infty$, as claimed. $\square$

134 **Proposition 6.** *Suppose that the assumptions of Theorem 5.1 hold. Then, with probability* $1$*, there*
135 *exists a (random) subsequence* $X_{n_k}$ *of (MD-b) which converges to Nash equilibrium.*

136 *Proof.* We begin with the technical observation that the set $\mathcal{X}^*$ of Nash equilibria of $\mathcal{G}$ is closed (and
137 hence, compact). Indeed, let $x_n^*$, $n = 1, 2, \ldots$, be a sequence of Nash equilibria converging to some
138 limit point $x^* \in \mathcal{X}$; to show that $\mathcal{X}^*$ is closed, it suffices to show that $x^* \in \mathcal{X}$. However, since Nash
139 equilibria of $\mathcal{G}$ satisfy the variational characterization (VI), we also have $\langle v(x), x - x_n^* \rangle \leq 0$ for all
140 $x \in \mathcal{X}$. Hence, with $x_n^* \to x^*$ as $n \to \infty$, it follows that

$$\langle v(x), x - x^* \rangle = \lim_{n \to \infty} \langle v(x), x - x_n^* \rangle \leq 0 \quad \text{for all } x \in \mathcal{X}, \tag{D.15}$$

141 i.e., $x^*$ satisfies (VI). Since $\mathcal{G}$ is monotone, we conclude that $x^*$ is a Nash equilibrium, as claimed.

142 Suppose now ad absurdum that, with positive probability, the pivot sequence $X_n$ generated by (MD-b)
143 admits no limit points in $\mathcal{X}^*$.[5] Conditioning on this event, and given that $\mathcal{X}^*$ is compact, there exists
144 a (nonempty) compact set $\mathcal{C} \subset \mathcal{X}$ such that $\mathcal{C} \cap \mathcal{X}^* = \varnothing$ and $X_n \in \mathcal{C}$ for all sufficiently large $n$.
145 Moreover, by (VI), we have $\langle v(x), x - x^* \rangle < 0$ whenever $x \in \mathcal{C}$ and $x^* \in \mathcal{X}^*$. Therefore, by the
146 continuity of $v$ and the compactness of $\mathcal{X}^*$ and $\mathcal{C}$, there exists some $c > 0$ such that

$$\langle v(x), x - x^* \rangle \leq -c \quad \text{for all } x \in \mathcal{C}, x^* \in \mathcal{X}. \tag{D.16}$$

147 To proceed, fix some $x^* \in \mathcal{X}^*$ and let $D_n = D(x^*, X_n)$ as in the proof of Proposition 5. Then,
148 telescoping (D.8) yields the estimate

$$D_{n+1} \leq D_1 + \sum_{k=1}^{n} \gamma_k \langle v(X_n), X_n - x^* \rangle + \sum_{k=1}^{n} \gamma_k \xi_{k+1} + \sum_{k=1}^{n} \gamma_k r_k + \sum_{k=1}^{n} \frac{\gamma_k^2}{2K} \|\hat{v}_n\|_*^2, \tag{D.17}$$

149 where, as in the proof of Proposition 5, we set

$$\xi_{n+1} = \langle U_{n+1}, X_n - x^* \rangle \tag{D.18}$$

150 and

$$r_n = \langle b_n, X_n - x^* \rangle. \tag{D.19}$$

151 Subsequently, letting $\tau_n = \sum_{k=1}^{n} \gamma_k$ and using (D.16), we obtain

$$D_{n+1} \leq D_1 - \tau_n \left[ c - \frac{\sum_{k=1}^{n} \gamma_k \xi_{k+1}}{\tau_n} - \frac{\sum_{k=1}^{n} \gamma_k r_k}{\tau_n} - \frac{(2K)^{-1} \sum_{k=1}^{n} \gamma_k^2 \|\hat{v}_k\|_*^2}{\tau_n} \right]. \tag{D.20}$$

152 Since $U_n$ is a martingale difference sequence with respect to $\mathcal{F}_n$, we have $\mathbb{E}[\xi_{n+1} \mid \mathcal{F}_n] = 0$ (recall
153 that $X_n$ is $\mathcal{F}_n$-measurable by construction). Moreover, by construction, there exists some constant
154 $\sigma > 0$ such that

$$\|U_{n+1}\|_*^2 \leq \frac{\sigma^2}{\delta_n^2}, \tag{D.21}$$

155 and hence:

$$\sum_{n=1}^{\infty} \gamma_n^2 \, \mathbb{E}[\xi_{n+1}^2 \mid \mathcal{F}_n] \leq \sum_{n=1}^{\infty} \gamma_n^2 \|X_n - x^*\|^2 \, \mathbb{E}[\|U_{n+1}\|_*^2 \mid \mathcal{F}_n]$$

$$\leq \text{diam}(\mathcal{X})^2 \sigma^2 \sum_{n=1}^{\infty} \frac{\gamma_n^2}{\delta_n^2} < \infty. \tag{D.22}$$

156 Therefore, by the law of large numbers for martingale difference sequences (Hall and Heyde, 1980,
157 Theorem 2.18), we conclude that $\tau_n^{-1} \sum_{k=1}^{n} \gamma_k \xi_{k+1}$ converges to $0$ with probability $1$.

158 For the third term in the brackets of (D.20) we have $r_n \to 0$ as $n \to \infty$ (a.s.). Since $\sum_{n=1}^{\infty} \gamma_n = \infty$,
159 it follows $\sum_{k=1}^{n} \gamma_k r_k / \sum_{k=1}^{n} \gamma_k \to 0$.

160 Finally, for the last term in the brackets of (D.20), let $S_{n+1} = \sum_{k=1}^{n} \gamma_k^2 \|\hat{v}_k\|_*^2$. Since $\hat{v}_k$ is $\mathcal{F}_n$-
161 measurable for all $k = 1, 2, \ldots, n-1$, we have

$$\mathbb{E}[S_{n+1} \mid \mathcal{F}_n] = \mathbb{E}\left[\sum_{k=1}^{n-1} \gamma_k^2 \|\hat{v}_k\|_*^2 + \gamma_n^2 \|\hat{v}_n\|_*^2 \,\middle|\, \mathcal{F}_n\right] = S_n + \gamma_n^2 \,\mathbb{E}[\|\hat{v}_n\|_*^2 \mid \mathcal{F}_n] \geq S_n, \quad \text{(D.23)}$$

162 i.e., $S_n$ is a submartingale with respect to $\mathcal{F}_n$. Furthermore, by the law of total expectation, we also
163 have

$$\mathbb{E}[S_{n+1}] = \mathbb{E}[\mathbb{E}[S_{n+1} \mid \mathcal{F}_n]] \leq V^2 \sum_{k=1}^{n} \frac{\gamma_k^2}{\delta_k^2} \leq V^2 \sum_{k=1}^{\infty} \frac{\gamma_k^2}{\delta_k^2} < \infty, \quad \text{(D.24)}$$

164 implying in turn that $S_n$ is uniformly bounded in $L^1$. Hence, by Doob's submartingale convergence
165 theorem (Hall and Heyde, 1980, Theorem 2.5), we conclude that $S_n$ converges to some (almost surely
166 finite) random variable $S_\infty$ with $\mathbb{E}[S_\infty] < \infty$. Consequently, we have $\lim_{n\to\infty} S_{n+1}/\tau_n = 0$ with
167 probability 1.

168 Applying all of the above to the estimate (D.20), we get $D_{n+1} \leq D_1 - c\tau_n/2$ for sufficiently large $n$,
169 and hence, $D(x^*, X_n) \to -\infty$, a contradiction. Going back to our original assumption, this shows
170 that at least one of the limit points of $X_n$ must lie in $\mathcal{X}^*$, so our proof is complete.  □

171 We are finally in a position to prove Theorem 5.1 regarding the convergence of (MD-b):

172 *Proof of Theorem 5.1.* By Proposition 6, there exists a (possibly random) Nash equilibrium $x^*$ of $\mathcal{G}$
173 such that $\|X_{n_k} - x^*\| \to 0$ for some (random) subsequence $X_{n_k}$. By the assumed reciprocity of the
174 Bregman divergence, this implies that $\liminf_{n\to\infty} D(x^*, X_n) = 0$ (a.s.). Since $\lim_{n\to\infty} D(x^*, X_n)$
175 exists with probability 1 (by Proposition 5), it follows that

$$\lim_{n\to\infty} D(x^*, X_n) = \liminf_{n\to\infty} D(x^*, X_n) = 0, \quad \text{(D.25)}$$

176 i.e., $X_n$ converges to $x^*$ by the first part of Proposition 3. Since $\delta_n \to 0$ and $\|\hat{X}_n - X_n\| = $
177 $\delta_n \|W_n\| = \mathcal{O}(\delta_n)$, our claim follows.  □

# E Rate of convergence

179 We now turn to the finite-time analysis of (MD-b). To begin, we briefly recall that a game $\mathcal{G}$ is
180 $\beta$-*strongly monotone* if it satisfies the condition

$$\sum_{i\in\mathcal{N}} \lambda_i \langle v_i(x') - v_i(x), x_i' - x_i \rangle \leq -\frac{\beta}{2} \|x - x'\|^2 \quad \text{($\beta$-DSC)}$$

181 for some $\lambda_i, \beta > 0$ and for all $x, x' \in \mathcal{X}$. Our aim in what follows will be to prove the following
182 convergence rate estimate for multi-agent mirror descent in strongly monotone games:

183 **Theorem 7.** *Let $x^*$ be the (unique) Nash equilibrium of a $\beta$-strongly monotone game. Then:*

184   a) *If the players have access to a gradient oracle satisfying (4.1) and they follow (MD) with*
185     *Euclidean projections and step-size sequence $\gamma_n = \gamma/n$ for some $\gamma > 1/\beta$, we have*

$$\mathbb{E}[\|X_n - x^*\|^2] = \mathcal{O}(n^{-1}). \quad \text{(E.1)}$$

186   b) *If the players only have bandit feedback and they follow (MD-b) with Euclidean projections*
187     *and parameters $\gamma_n = \gamma/n$ and $\delta_n = \delta/n^{1/3}$ with $\gamma > 1/(3\beta)$ and $\delta > 0$, we have*

$$\mathbb{E}[\|\hat{X}_n - x^*\|^2] = \mathcal{O}(n^{-1/3}). \quad \text{(E.2)}$$

188 *Remark.* Theorem 5.2 is recovered by the second part of Theorem 7 above; the first part (which was
189 alluded to in the main paper) serves as a benchmark to quantify the gap between bandit and oracle
190 feedback.

191 For the proof of Theorem 7 we will need the following lemma on numerical sequences, a version of
192 which is often attributed to Chung (1954):

193 **Lemma 8.** *Let $a_n$, $n = 1, 2, \ldots$, be a non-negative sequence such that*

$$a_{n+1} \le a_n \left(1 - \frac{P}{n^p}\right) + \frac{Q}{n^{p+q}} \tag{E.3}$$

194 *where $0 < p \le 1$, $q > 0$, and $P, Q > 0$. Then, assuming $P > q$ if $p = 1$, we have*

$$a_n \le \frac{Q}{R}\frac{1}{n^q} + o\left(\frac{1}{n^q}\right), \tag{E.4}$$

195 *with $R = P$ if $p < 1$ and $R = P - q$ if $p = 1$.*

196 *Proof.* Clearly, it suffices to show that $\limsup_{n\to\infty} n^q a_n \le Q/R$. To that end, write $q_n = n[(1 +$
197 $1/n)^q - 1]$, so $(1 + 1/n)^q = 1 + q_n/n$ and $q_n \to q$ as $n \to \infty$. Then, multiplying both sides of (E.3)
198 by $(n+1)^q$ and letting $\tilde{a}_n = a_n n^q$, we get

$$\begin{aligned} \tilde{a}_{n+1} &\le a_n (n+1)^q \left(1 - \frac{P}{n^p}\right) + \frac{Q(n+1)^q}{n^{p+q}} \\ &= \tilde{a}_n \left(1 + \frac{q_n}{n}\right)\left(1 - \frac{P}{n^p}\right) + \frac{Q(1 + q_n/n)}{n^p} \\ &= \tilde{a}_n \left[1 + \frac{q_n}{n} - \frac{P}{n^p} + \mathcal{O}\left(\frac{1}{n^{p+1}}\right)\right] + \frac{Q_n}{n^p}, \end{aligned} \tag{E.5}$$

199 where we set $Q_n = Q(1 + q_n/n)$, so $Q_n \to Q$ as $n \to \infty$. Then, under the assumption that $P > q$
200 when $p = 1$, (E.5) can be rewritten as

$$\tilde{a}_{n+1} \le \tilde{a}_n \left(1 - \frac{R_n}{n^p}\right) + \frac{Q_n}{n^p}, \tag{E.6}$$

201 for some sequence $R_n$ with $R_n \to R$ as $n \to \infty$.

202 Now, fix some small enough $\varepsilon > 0$. From (E.6), we readily get

$$\tilde{a}_{n+1} \le \tilde{a}_n - \frac{R_n \tilde{a}_n - Q_n}{n^p}. \tag{E.7}$$

203 Since $R_n \to R$ and $Q_n \to Q$ as $n \to \infty$, we will have $R_n > R - \varepsilon$ and $Q_n < Q + \varepsilon$ for all $n$
204 greater than some $n_\varepsilon$. Thus, if $n \ge n_\varepsilon$ and $(R - \varepsilon)\tilde{a}_n - (Q + \varepsilon) > \varepsilon$, we will also have

$$\tilde{a}_{n+1} \le \tilde{a}_n - \frac{R_n \tilde{a}_n - Q_n}{n^p} \le \tilde{a}_n - \frac{(R - \varepsilon)\tilde{a}_n - (Q + \varepsilon)}{n^p} \le \tilde{a}_n - \frac{\varepsilon}{n^p}. \tag{E.8}$$

205 The above shows that, as long as $\tilde{a}_n > (Q+2\varepsilon)/(R-\varepsilon)$, $\tilde{a}_n$ will decrease at least by $\varepsilon/n^p$ at each step.
206 In turn, since $\sum_{n=1}^{\infty}(1/n^p) = \infty$, it follows by telescoping that $\limsup_{n\to\infty} \tilde{a}_n \le (Q+2\varepsilon)/(R-\varepsilon)$.
207 Hence, with $\varepsilon$ arbitrary, we conclude that $\limsup_{n\to\infty} a_n n^q \le Q/R$, as claimed. $\qquad\square$

208 *Proof of Theorem 7.* We begin with the second part of the theorem; the first part will follow by
209 setting some estimates equal to zero, so the analysis is more streamlined that way. Also, as in the
210 previous section, we tacitly assume that ($\beta$-DSC) holds with weights $\lambda_i = 1$ for all $i \in \mathcal{N}$. If this
211 is not the case, the Bregman divergence $D(p, x)$ should be replaced by the weight-adjusted variant
212 (D.6), but this would only make the presentation more difficult to follow, so we omit the details.

213 The main component of our proof is the estimate (D.8), which, for convenience (and with notation as
214 in the previous section), we also reproduce below:

$$D_{n+1} \le D_n + \gamma_n \langle v(X_n), X_n - x^* \rangle + \gamma_n \xi_{n+1} + \gamma_n r_n + \frac{\gamma_n^2}{2K}\|\hat{v}_n\|_*^2. \tag{E.9}$$

215 In the above, since the algorithm is run with Euclidean projections, $D_n = \frac{1}{2}\|X_n - x^*\|^2$; other
216 than that, $\xi_n$ and $r_n$ are defined as in (D.18) and (D.19) respectively. Since the game is $\beta$-strongly
217 monotone and $x^*$ is a Nash equilibrium, we further have

$$\langle v(X_n), X_n - x^* \rangle \le \langle v(X_n) - v(x^*), X_n - x^* \rangle \le -\frac{\beta}{2}\|X_n - x^*\|^2 = -\beta D_n, \tag{E.10}$$

so (E.9) becomes

$$D_{n+1} \leq (1 - \beta\gamma_n)D_n + \gamma_n\xi_{n+1} + \gamma_n r_n + \frac{\gamma_n^2}{2K}\|\hat{v}_n\|_*^2. \tag{E.11}$$

Thus, letting $\bar{D}_n = \mathbb{E}[D_n]$ and taking expectations, we obtain

$$\bar{D}_{n+1} \leq (1 - \beta\gamma_n)\bar{D}_n + B\gamma_n\delta_n + \frac{V^2}{2K}\frac{\gamma_n^2}{\delta_n^2}, \tag{E.12}$$

with $B$ and $V$ defined as in the proof of Theorem 5.1 in the previous section.

Now, substituting $\gamma_n = \gamma/n^p$ and $\delta_n = \delta/n^q$ in (E.12) readily yields

$$\bar{D}_{n+1} \leq \left(1 - \frac{\beta\gamma}{n^p}\right)\bar{D}_n + \frac{B\gamma\delta}{n^{p+q}} + \frac{V^2\gamma^2\delta^2}{2Kn^{2(p-q)}}. \tag{E.13}$$

Hence, taking $p = 1$ and $q = 1/3$, the last two exponents are equated, leading to the estimate

$$\bar{D}_{n+1} \leq \left(1 - \frac{\beta\gamma}{n}\right)\bar{D}_n + \frac{C}{n^{4/3}}, \tag{E.14}$$

with $C = \gamma\delta B + (2K)^{-1}\gamma^2\delta^2 V^2$. Thus, with $\beta\gamma > 1/3$, applying Lemma 8 with $p = 1$ and $q = 1/3$, we finally obtain $\bar{D}_n = \mathcal{O}(1/n^{1/3})$.

The proof for the oracle case is similar: the key observation is that the bound (E.12) becomes

$$\bar{D}_{n+1} \leq (1 - \beta\gamma_n)\bar{D}_n + \frac{V^2}{2K}\gamma_n^2, \tag{E.15}$$

with $V$ defined as in (4.1). Hence, taking $\gamma_n = \gamma/n$ with $\beta\gamma > 1$ and applying again Lemma 8 with $p = q = 1$, we obtain $\bar{D}_n = \mathcal{O}(1/n)$ and our proof is complete. $\qquad\square$

To conclude, we note that the $\mathcal{O}(1/n^{1/3})$ bound of Theorem 7 cannot be readily improved by choosing a different step-size schedule of the form $\gamma_n \propto 1/n^p$ for some $p < 1$. Indeed, applying Lemma 8 to the estimate (E.13) yields a bound which is either $\mathcal{O}(1/n^q)$ or $\mathcal{O}(1/n^{p-2q})$, depending on which exponent is larger. Equating the two exponents (otherwise, one term would be slower than the other), we get $q = p/3$, leading again to a $\mathcal{O}(1/n^{1/3})$ bound. Unless one has finer control on the bias/variance of the SPSA gradient estimator used in (MD-b), we do not see a way of improving this bound in the current context.

## Footnotes

[1] We also recall here that $\mathcal{Y}$ comes naturally equipped with the dual norm $\|y\|_* = \max\{|\langle y, x \rangle| : \|x\| \le 1\}$.

[2]By "tangent" we mean here that $z$ belongs to the tangent cone $\mathrm{TC}(x)$ to $\mathcal{X}$ at $x$, i.e., the intersection of all supporting (closed) half-spaces of $\mathcal{X}$ at $x$.

[3] Recall here that $X_i^\delta$, $i \in \mathcal{N}$, denotes the $\delta$-adjusted pivot $X_i^\delta = X_i + r_i^{-1}\delta(X_i - p_i)$, i.e., including the feasibility adjustment $r_i^{-1}(X_i - p_i)$.

[4] In particular, this shows that $\mathbb{E}[D_n \,|\, \mathcal{F}_{n-1}]$ is quasi-Fejér in the sense of Combettes (2001).

[5] We assume here without loss of generality that $\mathcal{X}^* \neq \mathcal{X}$; otherwise, there is nothing to show.