[Reviews · NeurIPS 2018]

Reviewer 1



The paper examines the long term behaviour of learning with bandit feedback in non-cooperative concave games. This is an interesting application that I am not aware of being addressed from the bandit perspective before. The papers provides a clear introduction to and motivation of the problem. It is clearly written with sufficient references. There are also multiple examples of possible applications of the proposed algorithms. The main drawback of the paper is lack of any experimental results, which would make some of the discussed methods easier to understand in an intuitive way.

Reviewer 2



Context: It is a classic result that empirical frequencies of actions for players playing regret minimization algorithms converges to a coarse corr equilibrium. CCEs are not necessarily desirable solution concepts because they sometimes admit irrational behavior. For monotone games, it is known that the empirical frequencies converge converge to nash equilibrium for agents playing FTRL. Recently, Mertikopoulos et al proved that the sequence of plays for FTRL converges to nash for games -- they prove something more general that goes beyond concave potential games, in fact. This work considers that case when each agent can only observe bandit feedback. In such a setting, each agent does FTRL on some estimate of the gradient through a FKM-like one point estimator constructed by a random perturbation. The main result of the paper is that even with limited feedback the dynamics for monotone concave games converge to nash, and they do so at T^{-1/3} rate (for a "strict" variety of the problem). While the result was not known before, I do not find it surprising. The previous work established analogous rates and results when the agent does FTRL on unbiased stochastic gradients. Given this, FKM estimators can be seen as \delta-biased estimators of variance proportional to dimension. Therefore, it is reasonable to expect that choosing \delta appropriately should result in the same result with worse rates -- that same balance also shows up in standard convex bandit analysis. Even the proof in the appendix mirrors the same when compared to Mertikopoulos et al. - After the authors' comments. - While the convergence rate analysis in the aforementioned paper does not concern itself with the actual sequence but only the ergodic sequence, it is likely that this is because they don't employ (H4 in the same paper) that the gradients are Lipschitz - the present work does. In the same paper, for instance, with the assumption of H4, the authors have been able to prove convergence (but admittedly not the rate) of the actual sequence. With regards to the second point, the difference I see is that the previous results are based on regret analysis. But for \beta-strongly monotone games (~strong convexity), the argument must proceed through a recursive inequality of the template -- \Delta(T+1) < (1-\eps)\Deta(T)+ \gamma -- this paper does so. But this change is very expected since even first-order optimization algorithms for SC-functions proceeds through a similar argument. In this light, I'd like to retain my score. I'd encourage the authors to contrast their work with the previous in a manner that allows a clear comparison.

Reviewer 3



This paper presents a learning algorithm for participants in a strongly monotone convex game, such that if all participants use this learning algorithm, they converge to the unique Nash equilibrium of this game at a rate of O(n^{-1/3}) (i.e. at round n, each player is within O(n^{-1/3}) of their Nash equilibrium strategy). This extends a long line of work of understanding what sorts of equilibria learning strategies can/will converge to when used to play games (e.g. if everyone uses a low-regret strategy, the time-averaged play converges to a coarse correlated equilibrium). The learning algorithm they propose can be thought of as a modification of online mirror descent. The main technical difficulty here is that online mirror descent is a first-order method, requiring the gradient of the loss function at the queried points, but in this setting only zeroth-order feedback (the loss from performing some action) is provided. Moreover, unlike other settings where it is possible to approximate the gradient of some point by zeroth-order feedback of several points (even just two) in the neighborhood of this point, here each player only has control over their own action (and thus cannot guarantee they can query multiple closeby points in the overall action space). The trick the authors use to overcome this difficulty is called "simultaneous perturbation stochastic approximation", which allows one to get an "estimate" of a gradient at a point via a single zeroth-order query. The main idea is if one would like to query the gradient at some point x in R^d, they instead choose a random direction z_i in the unit d-sphere, and evaluate the function at x + delta*z_i. It can then be shown that f(x+delta*z_i)*d*z_i/delta is an unbiased estimator of the gradient of f~ at x, where f~ is some smoothed approximation to f. If f and its gradients are Lipschitz continuous, f~ gets closer and closer to f as delta goes to 0 (at the cost of the variance of this estimator exploding). Choosing delta to be something like n^{-1/3} optimizes this bias-variance tradeoff and leads to a convergence rate of O(n^{-1/3}). I think this is a strong result. Monotone concave games are a large class of interesting games, and a natural setting for this sort of question given their unique equilibrium structure. I really liked the application of the perturbation trick, which has appeared in the literature before, but which I had not seen and is applied nicely here. I have not checked the technical details in the supplemental material carefully, but the proof sketch makes sense to me. From a game theory perspective, it would be nice to have some slightly broader characterization of learning algorithms such that, if all players are running some learning algorithm in this class, then they converge to the Nash equilibrium of this game (at some convergence rate). This is a good first step in this direction, showing that there is some simple learning algorithm all learners can run to converge to a Nash equilbirium, but for the purpose of understanding how equilibria are achieved, it is unlikely that all players are running exactly this variant of mirror descent. I think it would be a very strong result if you could somehow show that if they are all using any sort of learning algorithm with "low X regret" (for some type of X regret), this convergence follows (such guarantees exist for convergence to CCE in general repeated games). The paper was written exceptionally clearly -- as someone who does not work closely with mirror descent, I really appreciated the detailed preliminaries in section 2 and 3. I recommend this paper for acceptance.